

# Levee system transformation in coevolution between human and water systems along the Kiso River, Japan

Shinichiro Nakamura[1], Fuko Nakai[2], Yuichiro Ito[2], Ginga Okada[2], Taikan Oki[3]

[1]Department of Civil and Environmental Engineering, Nagoya University, Nagoya, 464-8603, Japan
[2]Department of Civil and Environmental Engineering, Nagoya Institute of Technology, Nagoya, 466-8555, Japan
[3]Department of Civil Engineering, The University of Tokyo, Tokyo, 113-8656, Japan

*Correspondence to*: Shinichiro Nakamura (shinichiro@civil.nagoya-u.ac.jp)

**Abstract.** Floodplain societies decide whether to protect themselves against floods (fight), live with floods (adapt), or a
combination of the two. The formation of a levee system is an important factor in determining whether a society fight or adapt to flood; however, these factors have been considered as fixed boundaries in previous studies in human-flood interaction. We analyse a levee system transformation process covering the past century, from the indigenous ring-type levee system with floods to modern continuous levees against floods in the Kiso River basin, Japan, by applying a historical sociohydrological approach. The results show degradation processes of the indigenous levee system and traditional
communities alongside the installation of modern continuous levees, and a trade-off relationship was observed between the lengths of both. There are interactions between the levee systems and the human-water system through various water uses and different scale components, and the dynamics within the region are connected to external socioeconomic trends through the installed modern levees and institutions.

## 1 Introduction

Water-related phenomena have emerged alongside the interactions between humans and water in the Anthropocene (Savenije et al., 2014; Sivapalan et al., 2014). Understanding the coevolution between human and water dynamics resulting in the wide range of phenomena that arise in different places and different contexts globally is a major challenge for water and risk management research (Di Baldassarre et al., 2019; Sivapalan & Blöschl, 2015). On flood plains, this coevolution is a highly concerning phenomenon in the field of sociohydrology, which aims to understand the dynamics and coevolution of
coupled human-flood systems (Di Baldassarre, Kooy, et al., 2013; Pande & Sivapalan, 2016; Sivapalan et al., 2012). Throughout the history of human civilization, humans have settled in floodplains, and human-flood coevolution has occurred (Di Baldassarre, Kooy, et al., 2013; Hirabayashi et al., 2013; Jongman et al., 2012; Tanoue et al., 2016). In general, floodplain societies decide whether to try to mitigate the risks by defending themselves against floods ("fight"), to live with floods ("adapt"), or to enact any combination of the two (Di Baldassarre et al., 2015; Di Baldassarre, Kooy, et al., 2013; Luu
et al., 2022; Ruknul Ferdous et al., 2018). Whether and how societies fight flooding or adapt to flooding depend on society-



specific economic and technological possibilities (Ruknul Ferdous et al., 2018). Particularly, levees and levee systems, typical technologies designed to fight floods, are subject to defining whether a society fights or adapts. Previous studies focused on analysing levees and their impacts to understand and classify the immediate flood behaviours of societies/regions globally (Burton & Cutter, 2008; Di Baldassarre et al., 2009; Luu et al., 2022; Thanh et al., 2019; Tobin, 1995; White, 1942).

In nonstational changes of society (Milly et al., 2008; Wagener et al., 2010), infrastructure development, including levee development, tends to define the nature of society. Although Japan, the target area herein, is economically developed, and its society is shaped by multiple technologies currently, the non-Western cultural sphere, including those in Asia, has experienced dramatic historical changes due to modern technologies (Nakamura & Oki, 2018). The Japanese modern era (1868–present) started with the opening of Japan to the world (Gluck, 1985). Modern concepts and water technologies were

imported from Dutch engineers, and modern water management projects began in Japan. In the premodern Japanese society, flood risk management in each basin involved living with floods was conducted using indigenous discontinuous levees known as "*Kasumi-tei* (霞堤)" and "*Wajyū-tei* (輪中堤)" (ring-type discontinuous levees) and traditional community-based flood-mitigation activities by which societies adapted to floods (Takahashi, 1990) (Fig. 1). However, after modern concepts and water technologies were applied, the indigenous discontinuous levees were changed into modern continuous levees,

enabling a highly developed society and reducing the flood frequency through defences against floods, indicating that societies are now fighting floods. Thus, Japanese society has experienced dramatic changes and rapid modernization alongside the levee system transformation from an indigenous discontinuous levee system to a modern continuous levee system during the last century (Fig. 1); such human-water system paradigm shifts resulting from new technological choices have also emerged in other non-Western countries globally (Goh, 2019; Morita, 2016; Wesselink, 2016).

To understand the coevolution between human and flood systems in floodplains, it is important to capture the levee system transformation processes. A series of studies have been conducted on coupled human-flood systems in the sociohydrology field to develop a dynamic human-flood interaction model and assess the flood risks in two generic types of societies: "fighting" or "adapting" societies and "technological" or "green" societies (Di Baldassarre et al., 2015; Di Baldassarre, Viglione, et al., 2013). Sociohydrological spaces, geographical areas in a landscape with sociohydrological features that give

rise to the emergence of distinct interactions and dynamics between society and water, were proposed and applied to identify societal patterns/types (Ruknul Ferdous et al., 2018). Case studies that have used these societal or landscape divisions have enabled different cases to be compared globally, relating contextual particularities to the general patterns described by the conceptual model (Ciullo et al., 2017; Knighton et al., 2021; Perera & Nakamura, 2023; Sarmento Buarque et al., 2020; Schoppa et al., 2022; Shibata et al., 2022). However, each of these studies monistically assumed a fixed/time-invariant

society, and in this concept, the levee system is considered a fixed condition. Thus, the process of a dramatic shift in society from the "adapt" to "fight" modes due to a levee system transformation, as was experienced in Japan, or a mixture of the two modes, has not been considered. How have levee system transformations historically occurred in society? How has the levee system transformation process affected the dynamics of human-flood interactions and vice versa?





This study aims to identify the historical processes of levee system transformations over the last century and their impacts on human-flood dynamics in the Kiso River basin (Fig. 2), the most famous basin for the indigenous ring-type discontinuous levee system, the "Wajyū-tei", in Japan (Takeuchi & Shaw, 2008). The target of this study, the Nobi Plain in the Kiso River basin, has high topography in the east and low topography in the west in the plain region; the western part of the region, including the downstream areas of the Nagara and Ibi Rivers, has historically suffered extensive flooding (Koide, 1970). Therefore, in this area, the indigenous ring-type discontinuous levee system has been formed to protect the villages and farmlands from flooding; one past study pointed out that parts of this ring-type levee already existed in the 14th century (Ando, 1975). Unique independent communities were formed in each ring. Each of these communities had their own traditional community-based flood-fighting system to defend themselves against floods, and elevated houses called "Mizuya (水屋)" were equipped with boats for evacuation. To achieve our research goals, we detected and analysed the historical changes in the levee system configuration over the last century by digitizing historic maps, thus enabling spatial and quantitative analyses of historical landscape features in a historical sociohydrological way (Zlinszky & Timár, 2013). Quantitative trend and narrative analyses were conducted on several sociohydrological variables. Finally, we present a causal loop diagram to reveal the dynamics of human-flood interactions throughout the levee system transformation process.

**(a) Premodern period: Flood-adapted society with ring-type discontinuous levees**

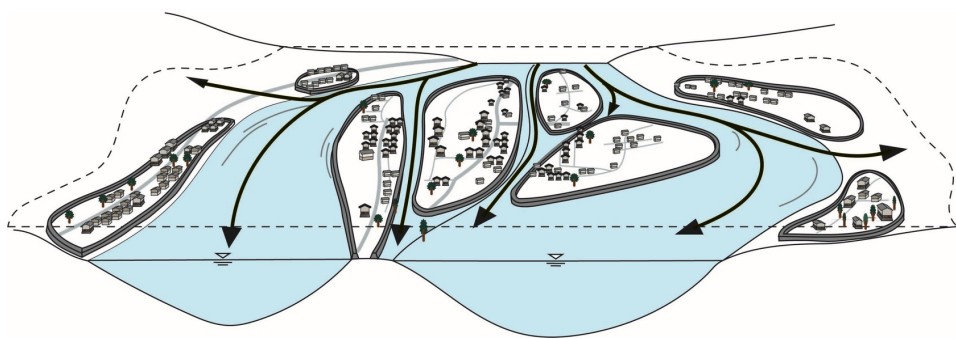

**(b) Present: Flood-fighting society with continuous levees**

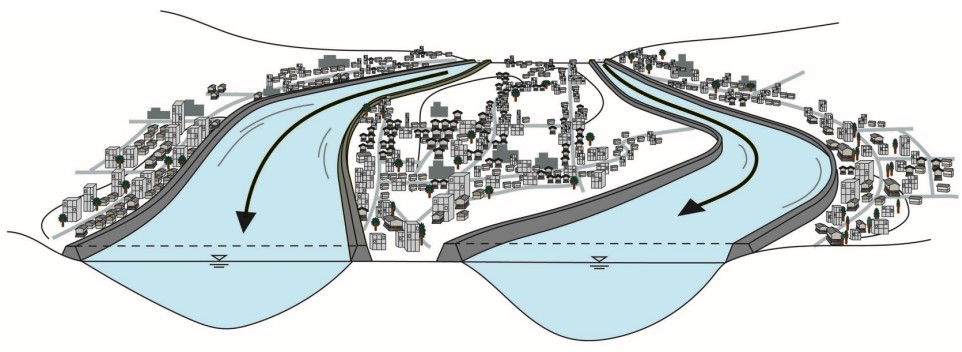





**Figure 1: Conceptual illustration showing the difference between the premodern and modern levee systems. The top panel (a) shows the indigenous ring-type discontinuous levee system in the premodern period, and the bottom panel (b) shows the modern continuous levees in present. Before modern times, each community was protected by an indigenous ring-type levee, and floodwaters flowed freely across the floodplain. Today, the floodplain is protected by modern continuous levees, and most of the ring levees have been lost. (Adapted from Di Baldassarre et al., 2015)**

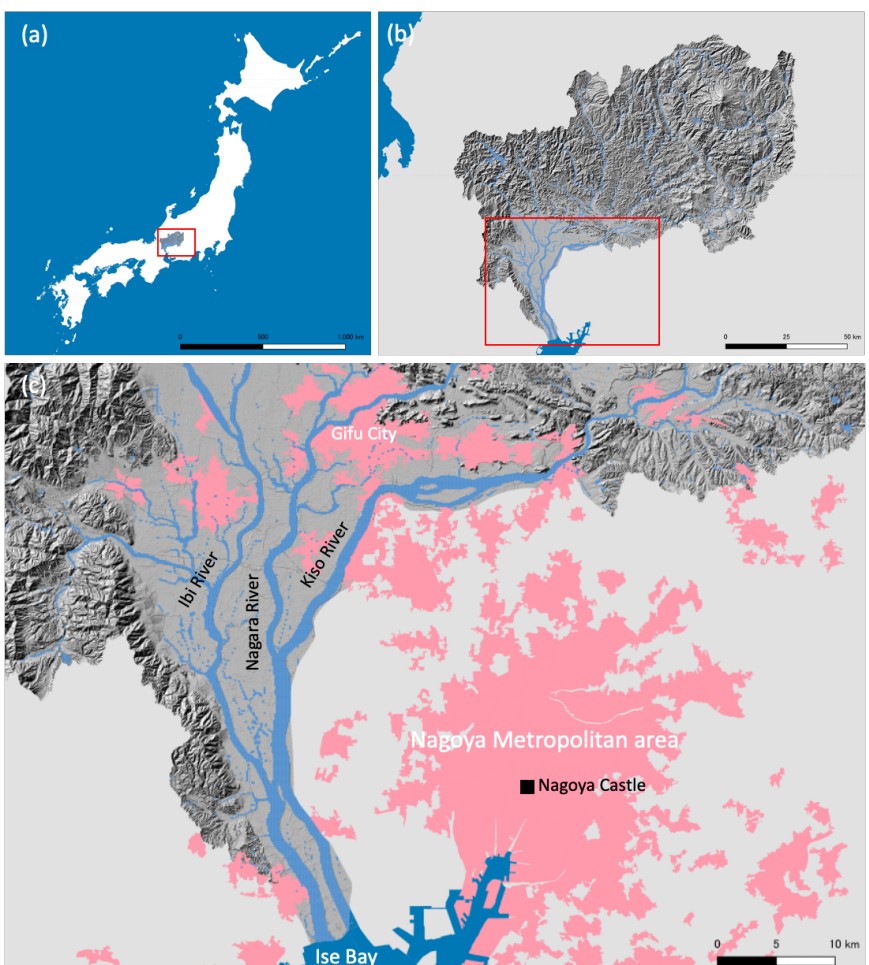

**Figure 2: (a) Japan as a whole, (b) the Kiso River basin, and (c) the Nobi Plain (the lower reaches of the Kiso River). The dark grey region indicates the areas considered in this study. The red areas represent densely populated areas. The Kiso River basin, the target basin of this study, is located in central Japan (a). The Kiso River basin consists of three subbasins, the Kiso River, Nagara River, and Ibi River basins, from east to west, and the total basin area is 9100 km2. These rivers formed the Nobi Plain (1485 km2), the second-largest plain in Japan, in which the Nagoya**





**metropolitan area, the third-largest metropolitan area in Japan (the population in the basin was 1.927 million in 2010), is located.**

## 2 Methods

### 2.1 Approach with historical sociohydrology

This study aims to identify the historical processes of levee system transformations and to show how this transformation historically impacted the dynamics of human-flood dynamics in the Kiso River basin, Japan. We took an interdisciplinary and historic approach to achieve this objective. Sociohydrology relies heavily on understanding historical processes to comprehend inherent fluctuations and how they have been addressed in the past (Kandasamy et al., 2014; Kim et al., 2021;
Nakamura & Oki, 2018; Wei et al., 2017; Yaeger et al., 2013; Zlinszky & Timár, 2013). Historical sociohydrology, a subdiscipline of sociohydrology that aims to understand a coupled system from its immediate or distant past, and its approaches must contribute to capturing paradigm shift processes throughout levee system transformations (Pande & Sivapalan, 2016; Sivapalan et al., 2012). This study applies historic methods; these methods have often been used to provide representative and reliable data given certain circumstances and are the main methods of historic hydrologic and
sociohydrological research (Zlinszky & Timár, 2013). In particular, historical maps provide a broad-scale representation of the landscape features associated with a given period. Therefore, systematic studies of hydrological changes based on historical maps have been conducted (Braga & Gervasoni, 1989; Bravard et al., 1986; Haidvogl et al., 2018; Hohensinner et al., 2013; Petts et al., 1989; Zlinszky & Timár, 2013).

### 2.2 Historical geographic dataset of traditional and modern levees and land use types

A historical geographic dataset of traditional and modern levees and land use types in the target area was created from historical maps representing multiple time periods to estimate the quantitative levee evolution. The target area, the downstream region of the Kiso River basin, including the Nobi Plain, is shown in Fig. 2 (c). We used two types of historic maps in the target area and various considered eras: maps from the Survey Bureau, Army Department, General Staff Headquarters, Imperial Japanese Army (SAGIJ maps) and maps from the Geospatial Information Authority of Japan (GIAJ
maps). The SAGIJ maps, the earliest modern maps in Japan with a scale of 1:20000, were prepared by the Imperial Japanese Army from 1888 to 1932 (Fig. S1). The survey used to construct the SAGIJ maps was conducted based on French and German measurement technologies. The GIAJ maps were prepared by the Geospatial Information Authority of Japan. The GIAJ maps, the most common topographic maps (with a scale of 1:25000) in Japan, have been generated since 1910. In this study, we digitized the maps at 8 eras in approximately 10-year intervals: the 1890s, 1920s, 1930s, 1940s 1960s, 1980s,
1990s, and 2000s. The list of maps is shown in the Supplemental Material (Table S1). The total number of map sheets was 133.



We digital-scanned and georeferenced the maps and clip the maps at the four corners corresponding to the latitude, longitude and control points (e.g., shrines and temples with unchanged locations) using the georeferenced tools of Arc GIS. The georeferenced historic maps were then compiled into a raster dataset representing each year. We visually determined and

traced the levee marks and converted the marks to shape data (Fig. S1). To differentiate indigenous ring-type levees and modern continuous levees, we referred to governmental reports from on-site surveys of the present levee locations. The estimated total length of modern continuous levees based on the map in 2007 was 411.9 km (the Kiso River basin contained 137.6 km, Nagara River basin contained 98.1 km, and Ibi River basin contained 176.2 km), and the official length of levees reported by the government in 2006 was 427.6 km. This result shows that the digitalized and estimated levee length data

have a high accuracy. To develop the land use dataset, 100-metre-mesh polygon data were created for each map representing each period using the Grid Index Features tool in Arc GIS. However, for the land use dataset, the interval was set to approximately every 20 years due to work-related resource limitations; the periods of the land use dataset are the 1890s, 1920s, 1940s, 1960s, 1980s, and 2000s. The land use attributes were assigned to each mesh by visually identifying the symbols on each map. When multiple land use types existed within a single mesh, the land use type judged to be the most

common within that mesh was adopted. The land use types were classified into eight categories: urban areas, public lands, factories, forests, fields, water areas, wastelands, and paddy fields. Finally, from the geometric data, we estimated the lengths of traditional and modern levees in each period, as well as the land use area.

## 2.3 Trend and narrative analysis

To quantitatively capture the phenomenon of changes in human-flood interactions due to the levee system transformation, a

trend analysis was conducted using the primary data obtained from the Japanese and prefectural governments and sociohydrological data independently generated from government reports and other sources. The time period covered by the trend analysis spanned 1885 to 2015 (130 years); however, the time period of each dataset was dependent on the data availability. Then, a narrative analysis referring to the trend analysis results was conducted using government reports and historical documents regarding the impacts of the levee system transformation on the dynamics of human-water interactions.

The key factors and relationships among factors are discussed and presented in a causal loop diagram regarding the impact of the levee system transformation on the human-flood system and vice versa. Previous sociohydrological studies have conceptualised the dynamics of human-flood interactions in floodplains in terms of five components, namely, the Flood, Technology, Demography, Society and Economy components; in addition, the interrelationships among these components have been conceptualised (Barendrecht et al., 2019; Di Baldassarre et al., 2015; Di Baldassarre, Viglione, et al., 2013). In

examining the causal loop diagram in this study, we considered extensions and improvements from the concept reported by Di Baldassarre et al. (2015) as a prototype.



## 3 Results

### 3.1 Process of levee system transformation

The evolution of the spatial distribution of levees (Fig. 3) clearly shows a gradual increase in the number of modern continuous levees and a gradual decrease in the number of indigenous levees. Over the past century, the length of modern continuous levees has nearly doubled, while the length of indfigenous discontinuous levees has been almost halved (Fig. 4 (a)). A trade-off has occurred between indigenous and modern levees, with the respective lengths of indigenous and modern levees reversing between 1947 and 1970 (Fig. 4 (a)).

The construction of modern continuous levees, which began in 1887, was already 56%, 41%, and 47% complete in the Kiso, Nagara, and Ibi River basins, respectively, and 48% complete overall in the 1890s (as of 1891) (Fig. 4 (a)). At this point, however, most of the river channels exhibited complicated meanders, and only a portion of the downstream sides had been straightened. During the beginning of the modern continuous levee construction period, the modern levee was developed by absorbing portions of the indigenous levees. This allowed for the efficient and rapid construction of modern levees: in the 1920s, levees were built between the Kiso and Nagara River basins to separate them, resulting in their channels being completely separated. In the 1920s, the modern levees reached the upstream regions, and 70% of the modern levees were completed. Ninety-two percent of the modern levees were completed in the 1970s, almost exhibiting their present forms (Fig. 4 (a)).

The indigenous levees existed with a total length of 612 km in the 1890s (as of 1891). The downstream indigenous levees were relatively large, with smaller indigenous levees upstream (Fig. 3). As time progressed, however, these indigenous levees were clearly lost as they were absorbed into or removed by modern continuous levees; as of 2007, the total length of indigenous levees was 293 km, only 48% of the 1890s length. In particular, the indigenous levee lengths declined substantially from the 1940s to the 1970s. The decrease during this 30-year period was approximately 30% of the total decrease. However, the decline of indigenous levees has clearly slowed since the 1980s.

Based on the levee system transformation process described above, its history can be divided into the following three phases.

Era 1 (1885-1945): indigenous levee systems dominated society

Era 2 (1945-1975): transition from a flood-adapted society to a flood-fighting society

Era 3 (1975-2015): stabilization of levee systems and society

The following sections provide trend and narrative analyses of the impact of each levee system transformation corresponding to each era on the mechanisms of human-flood interactions and vice versa. The summary of representative events and characteristics in each era are shown in Table 1.





**Figure 3: Levee system transformation and land use change processes in the Kiso River basin.**




**Table 1: Transition of events and trends regarding levee system, socio-economy, flood, water use in each era.**

| Events/trends | Era 1 (1885-1945) | Era 2 (1945-1975) | Era 3 (1975-2015) |
|---|---|---|---|
| **Levee system** | - Indigenous ring-type levee discontinuous system dominated<br>- Beginning of installation of modern continuous levees | - Transition from indigenous ring-type discontinuous levee system dominated to modern continuous levee system | - Modern continuous levee system dominated<br>- Stabilization of levee systems transformation. |
| **Socio-economy** | - *Meiji* Restoration<br>- Rapid modernization /industrialization<br>- World War I<br>- Population increase<br>- Food shortage | - World War II<br>- Rapid industrialization<br>- High economic growth<br>- Population increase<br>- Food shortage<br>- Electricity shortage | - Economic stagnation<br>- Changes in industrial structures from agriculture to manufacturing/service industries<br>- Population stabilization/decline |
| **Flood** | - Frequent floods<br>- Less flood damages<br>- Beginning of flood protection project with modern technologies | - Frequent sever floods<br>- Large flood damages<br>- Installation of dam reservoirs<br>- Decrease in number of flood fighters | - Less flood frequency<br>- The September 1976 flood (the *Anpachi* Flood)<br>- Decrease in number of flood fighters |
| **Water use** | - River channel improvement project for irrigation and navigation | - Water demands increase<br>- Sever water shortage<br>- Installation of dam reservoirs<br>- Paddy fields improvement<br>- Groundwater pumping increase<br>- Land subsidence | - Water demands stabilized<br>- Less water shortage<br>- Installation of dam reservoirs |

## 3.2 Levee system transformation and human-flood interactions

### 3.2.1 Era 1 (1885-1945): indigenous levee systems dominated society


Since 1608, a 48-kilometre-long continuous levee has been built along the left bank of the Kiso River to protect Nagoya Castle and the castle town. This embankment further intensified flooding in the area west of the right bank of the Kiso River. In 1754, the Edo shogunate (the government of Japan's early modern period) initiated a flood control program aimed at diverting three rivers, only a portion of which was completed (Kiso River Lower River Works Office, Chubu Regional

Development Bureau, Ministry of Construction, 1987). In the late 19th century, more than 80 ring levees still existed on the floodplain (Ando, 1975). The Japanese modern era (1868–present) started by the Meiji Restoration in 1868 with the opening of Japan to the world and the establishment of a modern government on behalf of the shogunate. Immediately after the opening of Japan to the world, Dutch engineers hired by the Japanese government began surveying rivers throughout Japan and studying ways to improve their management. Based on the results of this survey, Japan's first independent technical

manual was prepared by Cornelis Johannes van Doorn (1837-1906), a Dutch engineer, thus introducing the first modern science on water and rivers in Japan (Yamamoto, 1999). The project to improve Japan's major rivers into modern river





channels was then initiated. The main objective of this modern river channel improvement project was the development of navigation channels. At this time, the development of railroads as well as shipping had begun, and the development of transportation infrastructure was an important issue for the modernization of Japanese society. Although the population
density in the floodplain was low and flood damage was not extremely severe, flood protection projects were also undertaken on some rivers, mainly to protect agricultural lands (Nishikawa, 1969).

One of the first river improvement projects was implemented in the Kiso River basin in 1886. This improvement project had three objectives: flood protection, improving the drainage capacity of paddy fields, and improving shipping channels. The design floods (Kiso River: 7,350 m3/s, Nagara River and Ibigawa River: 4,170 m3/s) were established based on the flood
discharge values observed in the 1885 flood, just one year before the project started (Fig. 4 (b)), and the channel-straightening and continuous levee construction projects were initiated for these design flood conditions (Kiso River Lower River Works Office, Chubu Regional Development Bureau, Ministry of Construction, 1987). In 1891, the five years after the project started, 199 km of continuous levees were completed (Fig. 4 (a)). Between 1895 and 1920, the flood level of the Kiso River exceeded the design flood level five times (Fig. 4 (b)). However, the land use of the floodplain at that time consisted
mostly of paddy fields, with only a small urban area (Fig. 4 (e)), and the economic damage, except for agricultural damage, was not extensive. In addition, many ring levees remained, and the capacity of communities and individuals to mitigate damage from floods was relatively high.

After World War I (1914-1918), Japan experienced severe food shortages. Thus, strengthening food self-sufficiency became a key policy issue for the Japanese government. Additionally, throughout the war, modern industrialization and urban
population growth accelerated and, as serious flood damage occurred throughout Japan in 1917, 1918, and 1921, the Japanese government developed flood protection plans for 10 major river basins, including the Kiso River basin (Nishikawa, 1969). In the Kiso River basin, the design floods of the Kiso, Nagara, and Ibi Rivers were increased to 9,738 m3/s, 4,450 m3/s, and 3,340 m3/s, respectively, in 1921; at this time, each river channel was widened and straightened, and continuous levees were installed (Fig. 4 (b)). In addition, another flood in 1932 led to a third improvement project in 1936 involving the
construction of continuous levees, strengthening and widening of river channels throughout the basin. These improvement projects resulted in the completion of approximately 80% of the modern levees, thus forming the present continuous levee configuration (Fig. 1, Fig. 4 (a)). The flood inundation frequency was also greatly reduced: between 1921 and 1950, the flood level of the Kiso River exceeded its design flood level only three times (Fig. 4 (b)). Thus, as continuous levees were constructed and the flooding frequency decreased, the length of indigenous levees also gradually decreased (Fig. 4 (a)).

**3.2.2 Era 2 (1945-1975): transition from a flood-adapted society to a flood-fighting society**

World War II ended in 1945; from the third improvement project in 1936 until the end of this war, major river improvement projects were suspended because Japan was under a wartime regime. In the approximately 10 years following the end of the war, Japan experienced numerous historic floods, resulting in 12,456 people being killed or missing nationwide. In response to this severe flooding damage, the Japanese government began in earnest to introduce a new technology, dam reservoirs, in





conjunction with the river channel improvement projects that had previously included the construction of modern levees (Nakamura & Oki, 2018). In the Kiso River basin, the fourth project was initiated in 1947 and included the construction of multi-purpose dam reservoirs  such as the Maruyama Dam in the Kiso River (1956) and the Yokoyama Dam in the Ibi River (1964) (Ministry of Land, Infrastructure, Transport and Tourism (MLIT), 2006). After this project, the total dam storage capacity of the Kiso River basin reached 49,770 m3 (Fig. 4 (c)), and the design flood of the Kiso River was enhanced to

9,700 m3/s in 1954 (Fig. 4 (b)). Subsequently, the plan was partially revised in 1963 due to floods in the Nagara River in 1959 and 1960. Under this plan, the design floods of the Nagara and Ibi Rivers were increased to 7,500 m3/s and 3,850 m3/s, respectively (Fig. 4 (b)). This increased dam storage capacity and design floods greatly reduced the flood overflow frequency: between 1940 and 1980, no flood flows exceeding the design flood of the Kiso River occurred, and only 3 and 5 floods exceeded the design flood capacities of the Nagara and Ibigawa Rivers, respectively (Fig. 4 (b)).

After World War II, the population of Japan increased rapidly. The population of the Kiso River basin increased rapidly (Fig. 4 (d)), and the urban area within the floodplain also increased (Fig. 4 (e)). With this rapid population growth, increasing food production became an important political issue. In Japan, measures to increase agricultural production capacity, including water resource development and land improvement measures, were implemented. For water resource development, multipurpose dams were constructed throughout the country for flood control, agricultural and industrial water resources,

and hydropower generation. Seven dams were constructed and planned in the Kiso River basin (Fig. 4 (c)). For land improvement, paddy fields were reshaped, and roads were built to increase agricultural productivity. These changes led to a significant increase in agricultural production. Rice production in the Kiso River basin increased rapidly from 1.6 billion yen in 1960 to 5.8 billion yen in 1978 (Fig. 4 (c)). With the reshaping of rice paddies and the construction of roads on this plain, the indigenous levees formed in conjunction with the natural topography were rapidly removed or disconnected; even today,

many indigenous levees that were disconnected by roads remain, and mobile sluice gates have been installed at these removed points in case of flooding. The context influencing this phenomenon might have been the decreased need for indigenous levees due to the decreased flood frequency or decreased societal memory of floods. During this period, the length of modern levees and the length of indigenous levees in the subject area completely reversed.

In Japan, water resource development proceeded rapidly after World War II. However, Japan's rapid economic growth from

the late 1950s to early 1970s led to serious water shortages due to the rapid increase in agricultural production, rapid population growth, and increased demands for water in other sectors. Therefore, groundwater pumping increased in the studied basins, as the surface water resources were not sufficient. This groundwater pumping caused land subsidence over the entire Nobi Plain, and from 1960 to 1980, the land subsided 135 cm (Fig. 4 (g)). As a result, the area below mean sea level increased on the Nobi Plain: this area spanned more than 230 km2 in the late 1950s and 270 km2 (18.5% of the entire

Nobi Plain) in 1982 (Daito, 2015). Subsidence increases the risk of flooding: in Typhoon Vera in 1959, which caused 5,238 deaths, 310 km2 of the Nobi Plain, including the Nagoya metropolitan area, was inundated, of which 230 km2 was flooded for over two months. In response to this severe land subsidence, national and prefectural governments issued a series of



regulations prohibiting groundwater pumping after 1974 (Daito, 2015). Although these regulations have reduced the amount of groundwater pumping, the risk of flooding due to land subsidence continues to the present day.

### 3.2.3 Era 3 (1975-2015): stabilization of levee systems and society

Due to the introduction of modern levees and dammed reservoirs in the modern era, the Kiso River basin has become dramatically better protected against flooding, and the flood overflow frequency has been greatly reduced. However, the September 1976 flood (the Anpachi Flood) caused a major breach of the modern levees on the right bank of the Nagara River, inundating an area of 16.6 km2 and damaging approximately 7,000 houses (Ministry of Land, Infrastructure, Transport and Tourism, n.d.). However, in the areas where the indigenous levees remained in place, the indigenous levees prevented flooding and limited the expansion of the inundated area (Fig. 5). The 1976 flood was a turning point in the history of the levee system as it triggered a reevaluation of the value of the indigenous levee system in the Kiso River basin (Ito, 1980); after this event, the rate of decline of the indigenous levees slowed.

In Japan, there are traditional community-based flood-mitigation activities in which flood fighters in each community voluntarily take flood damage mitigation measures when floods occur (Yasuda, 1968). In the Kiso River basin, flood-mitigation systems existed primarily in each indigenous ring-type levee, and traditional community units were spatially defined by the indigenous ring-type levees. However, with the reduction and severance of indigenous levees in Era 2, the community units defined by the indigenous levees collapsed or were reorganized over relatively widespread areas (Ando, 1975). Furthermore, Japan's rapid economic growth that began in the 1950s decelerated in the late 1970s, and a period of economic stagnation began. Japan's rapid economic growth altered the lifestyle of Japanese communities based on traditional rice agriculture. In 1950, the primary industry accounted for 32.7% of all Japanese workers by industry, the secondary industry accounted for 29.1%, and the tertiary industry accounted for 38.2%. Through rapid economic growth, the primary industry share declined significantly, reaching 13.8% in 1975, while the secondary and tertiary industry shares rose to 34.1% and 51.8%, respectively (Ministry of Health, Labor and Welfare, 2013). With these industry shifts, the local communities and lifestyles based on the rice cultivation industry underwent major transformations to corporate employees in other industries, and as the construction of modern levees and dam reservoirs reduced the flooding frequency, the flood-fighting systems were weakened. In the Kiso River basin, the number of flood fighters decreased from 35,675 in 1964 to 20,956 in 2015 (Fig. 4 (h)); in Japan as a whole, the number of flood fighters decreased from 1.2 million in 1971 to 0.8 million in 2020 (Ministry of Land, Infrastructure, Transport and Tourism (MLIT), 2022).




295  .



**Figure 4: Historical evolution of the levee lengths and sociohydrological components in the Kiso River basin. The period covered is 1885-2015. However, the duration of each data period depends on the data availability. (a) Lengths of indigenous discontinuous and modern continuous levees, (b) annual maximum discharges and design floods of the**

300  **Kiso, Nagara, and Ibi Rivers, (c) total dam storage within the Kiso River basin (Misogawa, Agigawa, Maruyama, Iwaya, Ataki, Yokoyama, and Tokuyama dams), (d) population within the basin, (e) Paddy fields area and urban area within the basin, (f) rice production (Gifu Prefecture), (g) land subsidence (at the lowest point of the Kiso River, Kuwana City), and (h) number of flood fighters (Gifu Prefecture).**





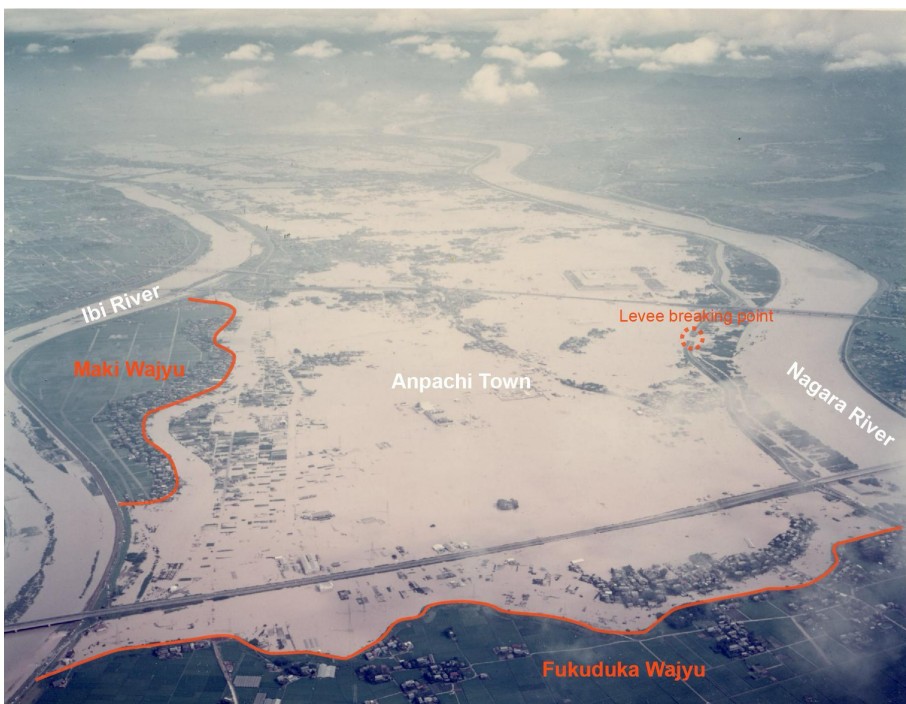

**Figure 5: September 1976 flooding. The Nagara River levee broke, and the floodwaters flowed out onto the floodplain, where Anpachi town was located. However, the floodwaters were prevented from expanding into the inundated area by the remaining ring levees (courtesy of the Ministry of Land, Infrastructure, Transport and Tourism).**

## 4 Discussions

Levees are an important factor defining human-flood systems and their dynamics. Through the approach of historical sociohydrology, in this work, we identified the levee system transformation process from an indigenous discontinuous levee system to a modern continuous levee system over the past century in the Kiso River Basin in Japan. We demonstrated the impact of this levee system transformation on the human-water system and its coevolution. The results reveal an interactive relationship between technologies and the human-flood system; the transformation of the levee system affects local communities and local culture, while social changes affect the local water management framework, including the levee system. With these interactions, Japanese society has shifted from adapting to and living with floods to fighting against them, thus characterising the levee system transformation process.

Here, we propose the relationship between a levee system transformation and the human-flood system represented as a causal loop diagram (Fig. 6). The causal loop diagram consists of the flood system, the water use system, and external social and economic trends. The flood system consists of "flooding", "memory", and "demography" components of previous studies, with the newly added "community mitigation (flood fighting)" and the trade-off relationship between the length of



"indigenous levees" and "modern levees". The memory positively affects the community mitigation and the indigenous levees extension. In the water use system, "demography" positively influences "water demand", and its increase accelerates groundwater extraction, which positively influences "land subsidence" and increased flooding. Increased water demand also promotes "dam reservoir storage": most dams built in postwar Japan were multi-purpose dams, which has a negative impact
on flooding. The socio-economic trends, such as changes in political regimes, wars, and economic growth, affect demography and the length of modern levees.

The levee system transformation in the Kiso River basin shows that there is a trade-off between modern continuous levees and indigenous levees. The construction of continuous levees began with the opening of Japan and the westernization of society, and their development was accelerated by social and hydrological drivers/trends such as flooding, war-induced food
shortages, industrialization, economic growth, and population growth/urbanization. The increased extension of continuous levees reduced the flooding frequency and could reduce people's memories of flooding (Di Baldassarre, Kooy, et al., 2013). These changes led to an increased population within the floodplain, a decreased mitigation capacity for floods (decrease in the number of flood fighters), and decreased needs for the indigenous levees. This process of decreased crises and opportunities for people to acquire the capacity necessary to deal with crises could be assumed as the high reliability
organization (HRO) principle (Dekker & Woods, 2010). On the other hand, the 1976 flood ended this downwards trend in the extension of ring levees, and this reminder of floods potentially triggered a reevaluation of indigenous technologies. In general, reevaluations of the versatility and flexibility of traditional or indigenous technologies have often emerged in the face of various water-related crises, including climate change (Mortimore & Adams, 2001; Pörtner et al., 2022; Zhu et al., 2020).

The increase in modern continuous levees reduced the flood frequency and memory of flooding and promoted population growth and development within the floodplain. Within the floodplain, increased water demands and inadequate surface water supplies led to increased groundwater pumping and land subsidence. This subsidence increased the flooding intensity, indicating that adaptation to drought conditions can exacerbate the negative impacts of flooding and vice versa (Di Baldassarre et al., 2018). This Kiso River basin case could be an example of a maladaptation to drought that increased the
risk of flooding through land subsidence. In Japan, the construction of a multipurpose dam in response to increased water demand successfully increased the water supply while simultaneously decreasing the flood frequency. This historical process is an important example of research on the combined mechanisms of drought and flooding or on water use and flood management (Baldassarre et al., 2017; Mazzoleni et al., 2021). Since the development of continuous levees was promoted in tandem with national politics and institutions, their developmental progress was influenced by external national or global
socioeconomic trends. Especially in studies of reservoir effects, researchers often explore the external socioeconomic trends affecting the internal system within a basin (floodplain) (Di Baldassarre et al., 2018; Godinez Madrigal et al., 2022); current human-flood coupled models take external trends into account by considering population growth rates. In the phenomena considered herein, external socioeconomic trends such as war and economic growth were also important factors that accelerated the development of continuous levees and changed the mode of the internal system. The result supports the



argument that connecting these different dynamic scales is an important theme in the previous sociohydrological studies (Vanelli et al., 2022; Yu et al., 2022). In recent years, there has been progress in the study of Global History within the field of historiography, aiming to elucidate processes that transcend regions, nations, and even singular civilizational boundaries (Conrad, 2016). The integration of this innovative approach in history and socio-hydrology also stands as a crucial challenge for the future.

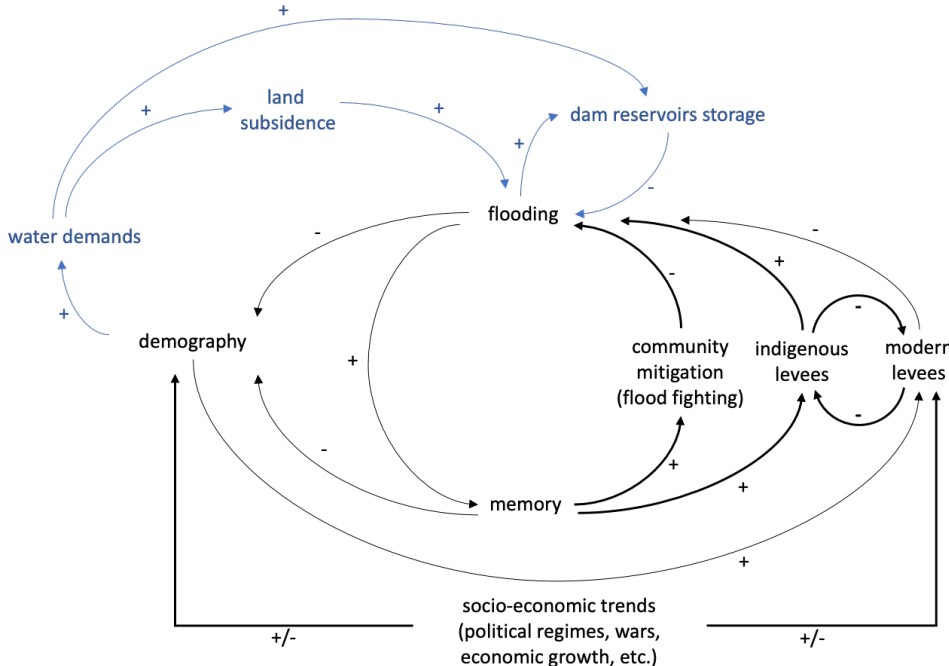

**Figure 6: Causal loop diagram of the levee system transformation and sociohydrological components, with black lines indicating systems related to humans and flooding, bold lines indicating the passes newly identified in this study versus in Di Baldassarre et al. (2015), and blue lines indicating feedback to flooding via the water use system. The positive and negative symbols indicate positive and negative impacts, respectively.**

## 5 Conclusion

This study revealed the process of levee system transformation from indigenous discontinuous levees to modern continuous levees in the Kiso River Basin, Japan, through the approach of historical sociohydrology. Over Japan's modernization in the past century, modern levees had been introduced, and with them, the indigenous levees declined: a trade-off relationship was observed between the two. Later, after the 1976 flood, the importance of indigenous levees was reevaluated and the removal of indigenous levees was stabilized, but the traditional community-based flood mitigation system that supports them is still deteriorating. The transformation caused a paradigm shift in society from "adapting to" to "fighting" floods with . In the process, these two different societal modes coexisted in one region, though the dominant society transitioned over time

alongside technological transformations. These changes dramatically transformed underdeveloped societies, resulting in rapid economic growth while simultaneously causing extreme changes in the original dynamics of human interactions with

water that, thus generating different challenges (increased social vulnerability and exposure).

This study emphasizes the need for water research to qualitatively and quantitatively observe the historical coevolution between human and water systems to truly understand related dynamics and processes. Previous sociohydrological studies have conceptualized the dynamics of human-flood interactions in floodplains in terms of related components. Although the mathematical representation of flood-loss processes in sociohydrological models has been criticized as overly simplistic

(Schoppa et al., 2022), these models are being applied and improved worldwide as the most reasonable method to understand human-water interaction phenomena. In this model, technology is one of the key components, considering the phenomenon that the already-existing levee height increases with each flood or that no levees exist at all. Thus, only different and independent trajectories can be represented: a society with levees or a society without levees. However, these models do not consider the significant changes that occur in levee systems, as shown in this study, and levees have been regarded as a fixed

boundary condition. Such drastic changes in levee systems due to the installation of continuous levees have been reported not only in Japan but also in many other Asian countries. Therefore, it could be necessary to discuss the impacts of levee system transformations on the dynamics of human-flood interactions, i.e., the relationship between "dynamic" technologies and other sociohydrological components. As such, it is important to co-design the models of the dynamics and processes together through discussions with social scientists. In particular, cooperation with history, which focuses on the historical

process of human-water interaction, could be highly beneficial.

**Data availability**

The levee length data generated in this study are available from the corresponding author upon reasonable request. Other relevant data are available from Japanese government agencies.

**Author contribution**

Contributions SN and TA conceptualized the study. SN curated the data, chose the methodology, performed the formal analysis and wrote the original draft. FN performed the formal analysis. GO and YI created dataset and wrote the original draft. All authors reviewed and edited the manuscript.

**Competing interests**

The contact authors have declared that none of the authors has any competing interests.



## Acknowledgments

This study was supported by JSPS KAKENHI (Grant Number: 18K13836 and 22H05234) and the JST-JICA-SATREPS
Program (JPMJSA1909).

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
