# Peer review of "Levee system transformation in coevolution between human and water systems along the Kiso River, Japan"

_EGUsphere, 2023_

## Author Response (AR1)

**Author's response**

**Levee system transformation in coevolution between human and water systems along the Kiso River, Japan**

Shinichiro Nakamura[1], Fuko Nakai[2], Yuichiro Ito[2], Ginga Okada[2], Taikan Oki[3]

[1]Department of Civil and Environmental Engineering, Nagoya University, Nagoya, 464-8603, Japan
[2]Department of Civil and Environmental Engineering, Nagoya Institute of Technology, Nagoya, 466-8555, Japan
[3]Department of Civil Engineering, The University of Tokyo, Tokyo, 113-8656, Japan

*Correspondence to*: Shinichiro Nakamura (shinichiro@civil.nagoya-u.ac.jp)

**#Reviewer 1**

**Comment 1**
Summary: Through historical sociohydrological lens, including analysis of historical maps, the authors present the how levee systems transitioned in Kiso river Japan from traditional to modern systems and present for the first time changing societies from green to technological societies as a function of exogenous, national level, economic conditions such as opening of national economics to international trade and knowledge.
Comments: this is an excellent paper, providing empirically grounded conceptual addition to existing causal loop diagram models of human flood system societies by introducing dynamic levee systems. These have been defined as fixed, either technological or green societies, in the state of the art models. Novel analysis of historical maps to identify "traditional" ring type dicontinuous levee systems and its transition to more linear modern levee systems is very interesting. The authors also nicely narrated based on historical accounts how societies with traditional system in the past were more living with the floods, while those in modern system seems to have forgotten it because modern levee system protected them against low magnitude high frequency floods only to be devastated by high magnitude rare flood events (even when compared to contemporary but less in number traditional ring levees). The only suggestion here would that perhaps the authors could focus their discussion a bit more on these novelties and highlight these novelties that the authors bring to the field of sociohydrology and perhaps whether ring levee system are more desirable even in industrialized economies such as Japan and if so what needs to be done to transition to such more living with nature systems.

**(Response)**
We sincerely appreciate your evaluation and the significant suggestions.

The desirable flood management system could be determined by circumstances in the target region such as the topographical characteristics, population trend and its spatial distribution, future flood intensity, and the abilities of maintaining the structures. As the results of this study showed, in the modern era, Japan has adopted a policy of continuous levees to protect flood plains, including rice paddies and farmlands, in order to cope with rapid population growth and food shortages. However, Japan is now in a phase of rapid population and economic decline, and in mountainous areas where continuous levees are not feasible in terms of investment potential and physical capacity, the policy has shifted to protecting only residential areas with ring levees, rejecting the use of continuous levees.

When adopting such flood management systems (including systems that live in harmony with nature), it is important to eliminate obstacles that limit equity participation in the discussion so as not to fall into the ineffective "deficit model" approach (Sullivan et. at., 2024) in the discussion process. And it is necessary to properly evaluate the benefits (cost-effectiveness, including ecological benefits) that flood management systems bring to the region, in coproduction with local communities, and to use the results of such evaluation to build consensus in the community.

Reference: Sullivan, J. A., H. K. Friedrich, B. Tellman, A. Saunders, and L. Belury (2024), Five key needs for addressing flood injustice, Eos, 105, https://doi.org/10.1029/2024EO240068. Published on 13 February 2024.

We added the above descriptions to L401-410 and Reference L530-531.

**#Reviewer 2**

This is a very comprehensive and interesting study to quantitatively summarize the socio-hydrological changes of a metro area in Japan. The data collection was comprehensive (with levees, land use/land cover change, socio-economic data) and with long time span (back to 1800s). The figures were carefully designed to clearly demonstrate the key messages. I only have a few moderate to minor suggestions.

**(Response)**

We sincerely appreciate your evaluation and suggestions. Below are our responses to your comments. We hope that our responses will satisfy your comments.

**(Comment 1)**

Regarding data availability:

Can the authors be more explicit in terms of "reasonable request"? Will levee geospatial data be shared? Will the levee length data be fully shareable as it is statistical attributes? Also, please provide the links for other relevant data.

**(Response)**

Thank you for your question regarding the data availability.

All data, including levee geospatial data, are available. However, some of the photos and data were obtained from the local governments, and their sources (including links) were listed in the supplementary Table S2.

**(Comment 2)**

Figure. 5: which one is the indigenous levee system? Is it Maki Waiyu? The main text does not mention the name and the figure does not highlight the levee, so it is quite confusing. Suggest authors to explicitly mark it on Figure 5, and also revise the texts to be more explicit (for international readers).

**(Response)**

Thank you for your appropriate indications.

The red lines in Figure 5 are the indigenous levee system. In the revised manuscript, we emphasized the lines in the caption of the Figure 5.

**(Comment 3)**

Figure 3: it would be clearer to directly mention data sources/resolution etc. in figure caption. This is an effective way to more clearly show the info without needing to go back to look for details in methods.

**(Response)**

We completely agree with your suggestion and added this figure information in the figure caption of Figure 3.

**(Comment 4)**

We also found in our study area the ring-type of levees are declining but they remained key in protecting unprecedented floods. They are perhaps the only left memory for a region haven't flooded for a while and is effective in its protection capability. Could the authors expand on these points?

**(Response)**

As the case of the Anpachi flood in this paper shows, the ring levees have the ability to reduce flood damages in the event of floods that exceed the capability of the continuous levees. However, in order to achieve this capability, it is necessary to maintain not only the physical/structural capacity of the levees, but also the operational capacity of the levees. In the target area, community flood fighting teams strengthen levees and open/close gates during floods, however as this paper shows, the flood fighting teams in the region are deteriorating. In order for the damage reduction ability of the ring levees to be effective, it is necessary to activate the activities for damage reduction in the local community.

We added the above description to L344-349.

**(Additional revision)**

- We deleted the reference of (Zlinszky & Timár, 2013) in L75. This was a correction of a simple mistake on our side.
- We added a funding which contributed for this revision to the acknowledgement.